# Tunable and high-purity room temperature single-photon emission from atomic defects in hexagonal boron nitride

Gabriele Grosso [1], Hyowon Moon [1], Benjamin Lienhard[1], Sajid Ali[2], Dmitri K. Efetov [1], Marco M. Furchi[3], Pablo Jarillo-Herrero[3], Michael J. Ford[2], Igor Aharonovich[2] & Dirk Englund[1]

Two-dimensional van der Waals materials have emerged as promising platforms for solid-state quantum information processing devices with unusual potential for heterogeneous assembly. Recently, bright and photostable single photon emitters were reported from atomic defects in layered hexagonal boron nitride (hBN), but controlling inhomogeneous spectral distribution and reducing multi-photon emission presented open challenges. Here, we demonstrate that strain control allows spectral tunability of hBN single photon emitters over 6 meV, and material processing sharply improves the single photon purity. We observe high single photon count rates exceeding $7 \times 10^6$ counts per second at saturation, after correcting for uncorrelated photon background. Furthermore, these emitters are stable to material transfer to other substrates. High-purity and photostable single photon emission at room temperature, together with spectral tunability and transferability, opens the door to scalable integration of high-quality quantum emitters in photonic quantum technologies.

[1] Department of Electrical Engineering and Computer Science, Massachusetts Institute of Technology, Cambridge, MA 02139, USA. [2] School of Mathematical and Physical Sciences, University of Technology Sydney, Ultimo, NSW 2007, Australia. [3] Department of Physics, Massachusetts Institute of Technology, Cambridge, MA 02139, USA. Gabriele Grosso and Hyowon Moon contributed equally to this work. Correspondence and requests for materials should be addressed to G.G. (email: ggrosso@mit.edu) or to D.E. (email: englund@mit.edu)

   1

Van der Waals materials allow for assembly of heterogeneous structures to realize new electronic, optical and materials functionalities[1, 2]. Recently, they have also emerged as promising materials for quantum information processing[3–5] with the discovery of stable quantum emitters in transition metal dichalcogenides (TMD)[6–10] and hexagonal boron nitride (hBN)[11–15]. hBN is a layered semiconductor with a wide band gap of 5.5 eV and has attracted considerable attention for its capability to enhance electronic and optical properties of two-dimensional (2D) material heterostructures[16], and for its natural hyperbolic properties[17]. In contrast to smaller band gap semiconductors, such as TMDs, where single photon emitters (SPEs) are attributed to excitons bound to impurities[6], emitters in hBN are associated with crystallographic defects, including the antisite defect $N_BV_N$ of the lattice, in which a nitrogen site is vacant and the neighbor boron atom is substituted by a nitrogen[13]. Atom-like defects in hBN confine electronic levels deep within the band gap and result in stable and extremely robust emitters with antibunched (i.e., non-classical) light. As recently reported, the emission energy of these emitters spans over a large spectral band[18], which presents a central problem for developing identical single photon sources. Furthermore, exfoliated hBN flakes show high background emission that reduces single photon purity.

In this article, we address these problems through strain control of the emission wavelength and a method to sharply reduce multi-photon emission probability, producing a tunable ultra-bright room temperature single photon source with the advantages of 2D materials, including stretchability, heterogeneous device assembly and straightforward integration with photonic circuits. The combination of photon purity, brightness, and tunability is essential for practical implementation of solid-state single photon sources in quantum information, quantum metrology and in situ light source characterization[4].

## Results

**Strain in two-dimensional materials.** Figure 1 illustrates the structure of a $N_BV_N$ defect in the hBN lattice, which is one of the possible atomic structures associated to the single-photon emission[13]. Photoluminescence (PL) measurements and second-order correlation function ($g^{(2)}(\tau)$), shown in Fig. 1a, characterize the emission of antibunched light emitted by point-like defects. The spectral tuning relies on strain: In 2D materials, due to the strong in-plane atomic bonds, external strain can be applied to alter the electronic energy levels of fluorescent defect states. The unusually high stretchability of 2D materials allows for effective strain engineering of physical and optical properties[19], including giant tunability of the electronic band gap. Recently, it has been shown that local strain in TMDs can be used to confine individual excitons with promising properties for single photon applications[20–22]. In the case of atom-like defects, the strain-induced displacement $\Delta a$ of lattice sites deforms the molecular orbitals and perturbs their energy levels according to $H_{strain} = \sum_{i,j} \sigma_{i,j}\varepsilon_{i,j}$, where $\varepsilon_{i,j} = \frac{\partial \delta_{x_i}}{\partial x_j}$ are the strain tensors and $\sigma_{i,j}$ are orbital operators. For hBN, the relevant strain directions are the zigzag (ZZ) and armchair (AC), indicated in Fig. 1b. The observable energy shift on the quantum emission can be therefore expressed as $E = E_{ZPL} + \sum_\Gamma \Delta E(\varepsilon)$, where $E_{ZPL}$ is the zero phonon line (ZPL) energy transition and $\Delta E(\varepsilon)$ is the energy shift of each orbital $\Gamma$ whose symmetry is broken by strain. A schematic of a simple case with a non-degenerate ground state $|g\rangle$ and one excited state $|e\rangle$ is shown in Fig. 1c. Like other solid-state systems[23, 24], external strain promises a particularly effective method to control the optical properties of embedded quantum emitters in 2D materials, as we will show below.

**Preparation and characterization of hBN samples.** The experiments used hBN samples prepared by a combination of focused ion beam (FIB) irradiation and high temperature annealing. We found that these steps can sharply reduce broad and bright autofluorescence of as-prepared hBN samples, which we ascribe primarily to organic surface residues[25], intrinsic defects[26] or impurities that create photoactive states within the band gap[27, 28] (more details in Supplementary Note 1). Figure 2 shows the effect of He$^+$ irradiation on an exfoliated 100 nm-thick hBN flake. This sample was irradiated in a $10 \times 10$ μm area with a

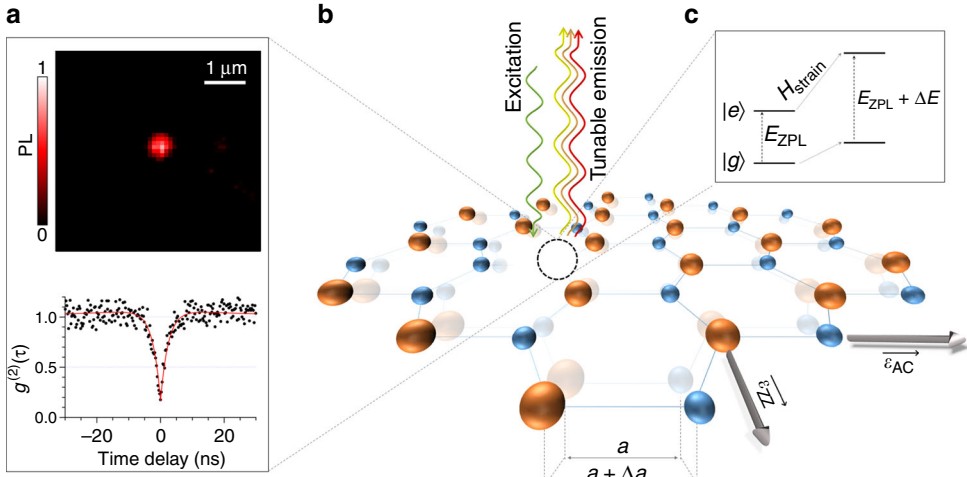

**Fig. 1** Single photon emission of atom-like defects in hexagonal boron nitride at room temperature. **a** Top image shows the normalized intensity of emission for an isolated single photon emitter in layered hBN when excited non-resonantly. The graph in the bottom is the second-correlation histogram of the quantum emission from the atom-like defect showing clear antibunching at zero time delay ($g^{(2)}(0) < 0.5$). The red line indicates the fitting as discussed in Supplementary Note 2. **b** Schematic representation of the hBN crystal lattice composed of regular hexagons with alternated atoms of N (*blue*) and B (*orange*). The *dotted black circle* shows an atom-like defect with a N atom missing and an adjacent B atom substituted by a N. Tensile strain along the armchair (AC) and zigzag (ZZ) directions produces a deformation of the crystal with a change $\Delta a$ of N-B bond length. Strain results in a shift of the energy levels of the atom-like defect and allows to tune the emission energy. **c** Drawing of a simplified energy level diagram showing the effect of strain. The emission energy can be strain-tuned of $\Delta E$ from the initial zero phonon line energy

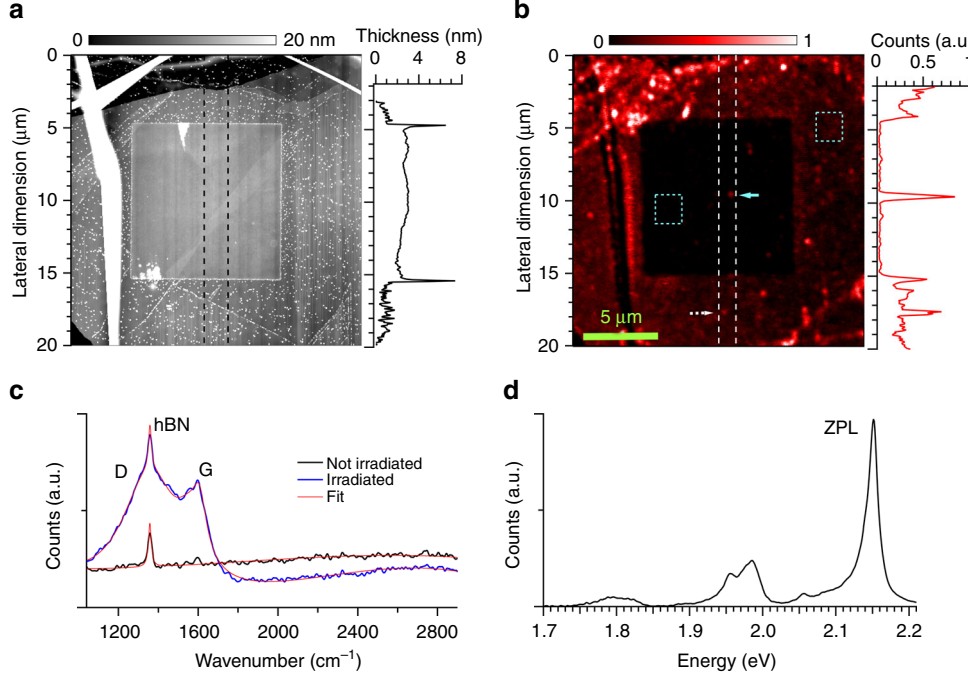

**Fig. 2** Spectroscopy of exfoliated hBN treated with focused ion beam. **a** Topographic map of the ion-irradiated area measured with atomic force microscopy. The hBN flake is ~100 nm thick and was irradiated with He ions with a dose of $5 \times 10^{15}$ ions per cm$^2$ in an area of 10 by 10 µm$^2$. The relative thickness is shown in the *left panel*. The profile is integrated over a 2 µm-large vertical area indicated with *dashed black lines* in the topographic map. The irradiated area presents a swelling of 1.2 nm with impurities accumulation of ~7 nm at the edges. **b** Confocal PL intensity map of the treated hBN flake in saturated and normalized color scale. The dark region shows a reduction of the background fluorescence with respect to the not irradiated area. Three single photon emitters can be identified within the irradiated region. The blue arrow indicates the emitter used for the photophysical studies. Emitters are also found outside the region (*white dashed arrow*). A vertical profile of the PL intensity (vertical *white dashed lines*) is illustrated in the *right panel* and it shows the increased visibility of the emitter inside the irradiated area due to the reduction of background emission. Two *blue squares* indicate the areas where Raman spectra are measured. **c** Raman spectra of the irradiated (*blue line*) and pristine region (*black line*). Both spectra are fitted with multi-peak functions to extract the Raman peaks (*red line*). In the irradiated area, the G- and D-bands appear on top of the Raman peak associated to hBN. **d** Room temperature spectrum of a single photon emitter inside the irradiated area (highlighted with a *blue arrow* in **b**). This emitter shows a zero phonon line emission at 2.145 eV with a spectral weight of ~30%

He$^+$ ion dose of $5 \times 10^{15}$ ions cm$^{-2}$ and successively annealed in Argon atmosphere at 1000 °C. The Atomic Force Microscopy (AFM) topography map of the sample after the treatment, shown in Fig. 2a, reveals far higher surface roughness in non-irradiated than irradiated regions, as well as a marked ~ 7 nm tall ridges at the borders between these regions. A vertical AFM profile shows that the thickness of the irradiated area is increased by approximately 1.2 nm. Such surface swelling has been ascribed to ion irradiation in other materials for similar ion dose: In Si[29] and diamond[30], swelling in crystal volume was attributed to amorphisation with migration and segregation of the displaced atoms from the bulk to the surface.

Our optical measurements suggest a more uniform hBN material in the irradiated region with a lower density of fluorescent defects. Figure 2b maps the photoluminescence intensity, acquired in a scanning confocal microscope with an excitation wavelength of 532 nm. The PL intensity of the irradiated region is reduced by five-fold compared to the non-irradiated area.

High resolution Raman spectroscopy in the non-irradiated area (black line in Fig. 2c) shows a single Raman peak around 1360 cm$^{-1}$ which is typically associated with hBN[31]. Inside the irradiated region (blue line), two broad peaks appear around 1355 cm$^{-1}$ and 1610 cm$^{-1}$, and are attributed to the D- and G-band, indicating a partial graphitization of the hBN surface[32, 33]. The absence of the Raman peak at 1290 cm$^{-1}$ associated to cBN allows us to exclude the possibility of the hBN-to-cBN transition[31, 33].

The reduced PL background in the irradiated region in Fig. 2b reveals three bright spots with sharply increased

signal-to-background ratio, as shown in the vertical profile of Fig. 2b. The spectrum at room temperature of an emitter inside the FIB area (blue arrow), seen in Fig. 2d, shows a ZPL emission around 2.145 eV and overall spectral matching with emitters previously associated to the antisite nitrogen vacancy (N$_B$V$_N$) type defects[11, 18].

**Photophysics of single photon emitters**. Figure 3a plots the second-order correlation histogram ($g^{(2)}(\tau)$) of a typical emitter from the irradiated area, as measured integrating over the entire emission spectrum and using a Hanbury Brown and Twiss setup with pulsed excitation of 300 ps and 20 MHz repetition rate. This measurement yields a single photon purity (multi-photon probability) of $g^{(2)}(0) = 0.077$. For comparison, Fig. 3b shows the same measurement for an emitter with a similar spectrum, but in the non-irradiated region with high background emission as indicated by the gray area in the inset, which produces much higher multi-photon probability, with $g^{(2)}(0) = 0.263$. This increase in $g^{(2)}(0)$ matches well with expectations for high values of uncorrelated background photon emission according to the following[34]:  $g_{exp}^{(2)}(0) = \rho^2(g_{ideal}^{(2)}(0) - 1) + 1 \approx 0.306$,    where $g_{exp}^{(2)}(0)$ is the expected value, $g_{ideal}^{(2)}(0) = 0$ for ideal emitters, $\rho = S/(S + B) \approx 0.83$, $S \approx 110$ and $B \approx 22$ are the emission intensity from the SPE and the background, respectively.

With background reduced, the emitters exhibited remarkably bright single photon emission. Figure 3c plots the rate of collected photons from a single hBN emitter as a function of continuous

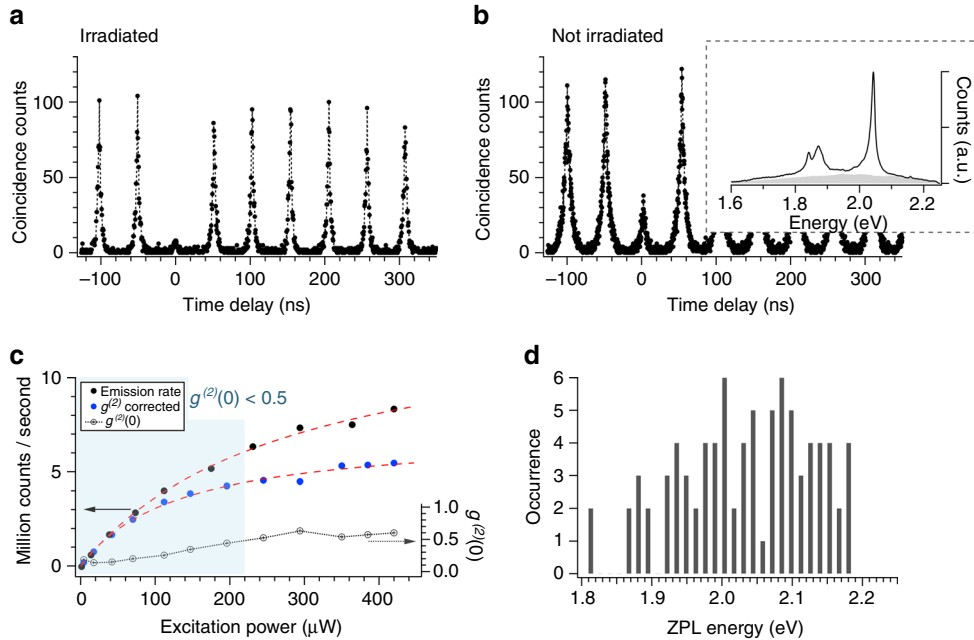

**Fig. 3** Photophysics of single photon emitters. **a** Autocorrelation histogram with pulsed excitation of 300 ps and 20 MHz repetition rate of the emitter inside the irradiated region, highlighted with the *blue arrow* in Fig. 2b. Fitting yields an emission purity of $g^{(2)}(0) = 0.077$. Black dots show the data points while the *dashed line* serves to guide the eye. **b** For comparison the autocorrelation function of an emitter outside the irradiated region but with similar spectrum is measured in the same experimental conditions. The single photon purity is drastically degraded to $g^{(2)}(0) = 0.263$ due to the background emission. *Black dots* show the data points while the dashed line serves to guide the eye. The inset shows the spectrum of the emitter outside the irradiated region and the *gray* area indicates the experimental background PL. **c** Measured emission rate (*black dots*) and emission rate corrected for unperfected $g^{(2)}(0)$ (*blue dots*) as a function of the excitation power. The *dashed red lines* are the result of the fitting function $I = I_{inf} \times P/(P + P_{sat})$, yielding a saturation intensity of $13.8 \pm 0.6 \times 10^6$ counts per second for the measured rate and $7.1 \pm 0.3 \times 10^6$ counts per second for the corrected rate. *Open circles* show the autocorrelation function at zero time delay $g^{(2)}(0)$ for CW excitation as a function of excitation power. The *light blue* area indicates the emission rates with antibunching ($g^{(2)}(0) < 0.5$). Single photon emission is maintained up to a rate emission above $6 \times 10^6$ counts per second. **d** The histogram shows the spectral distribution of the zero phonon line emission for approximately 90 emitters with a bin size of 10 meV

wave (CW) excitation power. This curve follows a saturated-emitter model for a single photon source reaching half its saturated emission intensity, $I_{inf}$, at a pump power $P_{sat}$: $I = I_{inf} \times P/(P + P_{sat})$. We neglected in this model the low measured background emission which is less than 4% of the total emission at maximum power, as shown in Supplementary Fig. 9. The fitting yields $I_{inf} = 13.8 \pm 0.6 \times 10^6$ counts s$^{-1}$ and $P_{sat} = 278$ μW. Power-dependent measurements of the autocorrelation function with continuous wave excitation (open circles in Fig. 3c) yield single-emitter antibunching $g^{(2)}(0) < 0.5$ even beyond $6 \times 10^6$ counts per second, but show a degradation as the pump power increases. We take into account the uncorrelated photon emission and correct the emission rate for imperfect $g^{(2)}(0)$ as follows[34]: $S = (S + B)\sqrt{1 - g_{exp}^{(2)}(0)}$. The corrected emission rate is shown in blue dots in Fig. 3c for which the fitting yields $I_{inf} = 7.1 \pm 0.3 \times 10^6$ counts per second and $P_{sat} = 131$ μW, making this emitter one of the brightest solid-state source at room temperature[4]. Measurements of the quantum efficiency are beyond the scope of this work and comparisons with other efficient systems, including organic molecules[35], are not possible. Time-dependent measurements revealed a lifetime of 2.4 ns for this emitter (see Supplementary Note 2 and Supplementary Fig. 10), in excellent agreement with previous measurements[13, 18].

**Strain tuning of quantum emission.** The studied emitters exhibit a large spectral distribution in the ZPLs as shown in Fig. 3d, in agreement with previous reports[18]. The wide span of the emission

energy has been theoretically attributed to the effect of local strain fluctuations in hBN layers, though experimental evidence was missing. To investigate this question, we transferred hBN films, after irradiation and annealing, onto a bendable 1.5 mm-thick polycarbonate (PC) beam that allowed us to controllably apply strain, confirming also that quantum emitters persisted this transfer process (see Supplementary Note 3). As illustrated in Fig. 4a, one edge of the PC beam was fixed while the other was bent downward (upward) to apply tensile (compressive) strain proportional to the beam deflection δ: $\varepsilon = 3h\delta(L - d)/(2L^3)$, where $h$ is the thickness of the beam, $d$ and $L$ are the distances between the clamp and the hBN sample and the deflection point, respectively. Polycarbonate has high Poisson's ratio of 0.37 that results in a weak compression along the width direction when tensile strain is applied along the length direction, and vice versa.

Figure 4b plots the ZPL shift for three different emitters as a function of applied strain. Linear fits indicate ZPL strain dependencies ranging from −3 to 6 meV/%. Figure 4c illustrates the emission spectra of the emitters with tunability 6 meV/% and −3.1 meV/%. These tuning coefficients have to be corrected for the experimental conditions. In the experiments, we used thick hBN flakes (~ 50–100 nm) and the strain amplitude at the emitter can be smaller than the applied strain on the surface of the bendable substrate due to the slippage among the layers composing the thick hBN flakes. Recent simulations[36] on bilayer van der Waals heterostructures show that, before a critical strain value, the strain transferred among two consecutive layers is $t \sim 98\%$. We assume that in thicker flakes the correction factor for

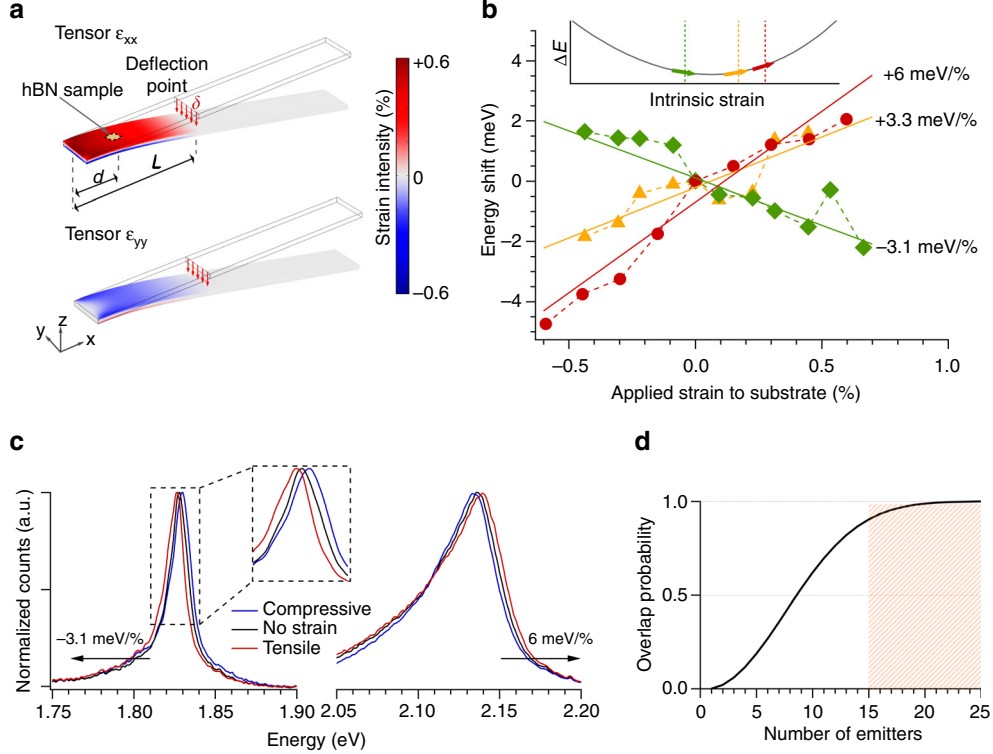

**Fig 4** Strain tuning of single photon emission. **a** Experimental scheme used to apply strain to hBN flakes sitting onto a bendable polycarbonate (PC) beam clamped at one edge. Colors show the simulated strain intensity along the length (*top panel*) and along the width (*bottom panel*) of the beam when a vertical force is applied to the free edge. When tensile strain is induced in the length direction (*x* axis), compressive strain occurs along the beam width (*y* axis) due to the PC Poisson's ratio of 0.37. hBN samples are transferred at a distance *d* from the clamp and controllable strain is applied when the PC beam is displaced by *δ* from its rest position at a distance *L* from the clamp (*red arrows*). **b** The plot shows the scaled energy shift as a function of applied strain to the bendable substrate for three emitters with different tunability of −3.1 meV/% (*green*), + 3.3 meV/% (*yellow*) and + 6 meV/% (*red*). Inset shows a sketch of a quadratic energy shift ΔE for the single photon emission induced by intrinsic strain. Different tunability is due to the different initial strain conditions at which the emitters are found at the beginning of the experiments, as indicated by matching *colored arrows* and *dashed vertical lines*. **c** Spectra of the emitters with tunability −3.1 meV/% and 6 meV/% for compressive (*blue curve*), tensile (*red curve*) and no strain (*black curve*). The values of the strain intensity are the following. *Left panel* −0.4%, 0%, + 0.4%. *Right panel* −0.6%, 0%, + 0.6%. **d** Probability that two of total *n* emitters can be spectrally overlapped as a function of increasing number of total emitters in the system considering a maximum tuning of 6 meV and a zero phonon line (ZPL) spectral distribution of 300 meV. Shadow area shows that this probability is higher than 90% when at least 15 emitters with random ZPL energy are found in the system

transmitted strain *t* is the same for all consecutive layers. In these conditions, the effective strain at the emitter location ($S_{eff}$) is given by the strain applied to the bottom layer of the hBN flake ($S_{app}$) scaled by the transmitted strain *t* among the *N* layers separating the emitter and the substrate, and reads $S_{eff}^N = S_{eff}^{N-1} \cdot t = \left(S_{eff}^{N-2} \cdot t\right) \cdot t = \ldots = S_{eff}^1 \cdot t^{N-1} = S_{app} \cdot t^N$. Considering an average of $N \sim 150$ layers for our experiments, we obtain $S_{eff} = S_{app} \cdot t^N = S_{app} \cdot 0.05$. Therefore, the amplitude of the tuning coefficients is underestimated by at least an order of magnitude. Moreover, the strain at the emitter also depends on the initial conditions of the flakes that initially are not at zero-strain because of wrinkles, cracks, and lattice mismatches caused by sample fabrication. These create random fluctuations in built-in strain of hBN flakes that are responsible of the large variation of the measured tuning coefficients as illustrated in the inset of Fig. 4b, in which the three different coefficients are schematically indicated with matching color arrows. The assumption of non-linear and non-monotonic energy shift is supported by theoretical simulations as discussed later. The exceptional elasticity of two-dimensional materials suggests that the tuning of SPEs in hBN is a reversible process within the strain range used in our experiments[37–40].

The tunable energy of SPEs in hBN can greatly increase the probability that multiple emitters have matching ZPL emission wavelengths. Considering the ZPL spectral distribution of ~ 300 meV, the probability that at least two emitters among a total of *n* emitters are found within a range of 6 meV (accessible by strain tuning) reads $P = 1 - \prod_{i=0}^{n-1}(1 - i \cdot p)$, where $p = 1/50$ corresponds to the probability that two random emitters fall in a 6 meV range bin. This probability as a function of *n* is calculated in Fig. 4d and shows that for just 15 emitters, at least two emitters can be spectrally matched with probability $P > 0.9$.

We used density functional theory in the Perdew Burke Ernzerhof approximation[41] (see Methods) to model the strain-induced energy shift of the hBN quantum emitters. These simulations considered four strain directions along the plane of the hBN sample, as labeled by the lattice directions shown in the inset of Fig. 5. A realistic case, similar to our experimental conditions, is taken into account by considering the effects of the Poisson's ratio of 0.37 of the PC substrate. In these conditions, tensile strain along AC2 produces a compression along the orthogonal direction ZZ1: $\varepsilon_{ZZ1} = -0.37 \times \varepsilon_{AC2}$, and vice versa. Figure 5 shows the optical response in the form of the imaginary part of the dielectric function ($\xi_2$) for tensile strain applied from 1 to 5% along ZZ1 (shown in blue tones) and along AC2 (shown in red tones). The peak around 2.01 eV (zero strain) is due to a transition between $N_B V_N$ defect levels that fall within the band gap with an electric field vector parallel to the in-plane B-N

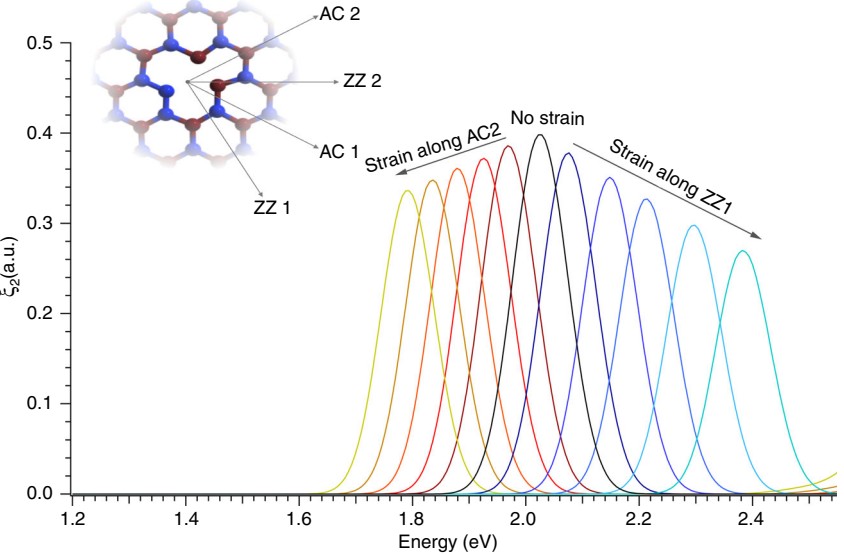

**Fig. 5** Simulated spectral distribution as a function of strain. The cartoon in the inset shows the lattice directions in the plane of the hexagonal lattice with respect to $N_BV_N$ defect. Four independent directions can be identified: two along the zigzag direction, ZZ1 and ZZ2, and two along the armchair direction, AC1 and AC2. The main plot is the simulated optical response in the form of the imaginary part of the dielectric function ($\xi_2$) as a function of strain applied in either zigzag or armchair direction. The simulation takes into account the effect of the high Poisson's ratio of the PC substrate of 0.37 that induces a compression (stretching) in the orthogonal direction with respect to the direction where tensile (compressive) strain is applied. The *black graph* shows the imaginary part of the dielectric function at zero strain. The graphs in *red* (*blue*) tones correspond to tensile strain applied along the AC2 (ZZ1) direction from 1 to 5%. In these conditions, the Poisson's ratio of the PC substrate results in a compression in the orthogonal direction from −0.37 to −1.85%

bonds. Our simulations confirm the non-monotonic behavior of the ZPL energy under the effect of strain and its role in the large spectral distribution observed experimentally. The simulated energy shifts due to strain along the other directions are reported in Supplementary Note 3.

In summary, we presented a technique to isolate bright emitters in exfoliated hBN and demonstrated a method to tune their spectral properties. The possibility of spectral tuning up to 6 meV greatly increases the probability of spectral overlapping among different emitters. We reported an emission rate near $10^7$ counts per second with high purity of $g^{(2)}(0) = 0.077$. These emitters are stable to transfer onto other substrates, which opens the possibility for integrating them into hybrid devices, such as photonic integrated circuits. The availability of bright, high-purity, spectrally tunable, and mechanically transferable SPEs shows strong promise of 2D materials for future semiconductor quantum information processing technologies.

## Methods

**Sample fabrication**. hBN flakes are mechanically exfoliated from bulk material onto a 285 nm $SiO_2$ substrate thermally grown on doped Si. After exfoliation, hBN flakes are irradiated with $He^+$ by using an ion microscope (Zeiss ORION NanoFab) which allows to focus the ion beam with a precision in the range of nanometers without the use of auxiliary masks. The beam current is set to $I_{beam} = 0.8$ pA and the acceleration to 32 keV with a gas pressure of $P \sim 5 \times 10^{-6}$ Torr. The ion dose is controlled by the dwell time (from 0.1 up to 500 μs), the irradiation area and spacing (from 256 pixels up to 2048 pixels). After irradiation, the flakes are annealed in Argon environment at 1000 °C for 30 min. For strain experiments the hBN flakes are wet transferred onto a 1.5-mm-thick polycarbonate (PC) beam with dimensions of 1.2 cm and 10 cm. Poly-methylmethacrylate (PMMA) is spin coated on the sacrificial Si/SiO2 substrate hosting the flakes at 1000 r.p.m. for 45 s followed by 2000 r.p.m. for 5 s. PMMA is dried in ambient conditions for 15 min and then immersed in DI water at 60 °C for approximately 2 h and mechanically lifted off the sacrificial Si/SiO2 substrate. Subsequently, the PMMA layer carrying the hBN flakes is transferred onto the PC beam (to increase the adhesion the PC is pre-cleaned by air plasma). As the thin

PMMA layer prevents slippage when strain is applied, it is not removed throughout the entire experiment.

**Optical characterization**. SPEs are characterized in a fiber-coupled confocal microscope. A schematic of the optic setup is shown in Supplementary Fig. 4. A green laser (532 nm) is used for excitation and PL maps are taken with a galvanometer mirror scanner in a 4f configuration. An oil-immersion microscope objective with NA = 1.42 and free-space avalanche photodiodes (APDs) are used for high efficiency collection. Spectroscopy of SPEs and Raman is done with a microscope objective with NA=1 and fiber-coupled spectrometer with either 100 or 300 lines per mm grating. In detection, the excitation laser is filtered out with a polarized beam splitter, a dichroic mirror and a 550-nm long-pass filter. The emission rate of the SPE is estimated considering a correction factor due to the efficiency and the module dead time of the APDs.

Second-order correlation histogram is measured integrating over the entire emission spectrum and without any spectral filtering, except for the 550-nm long-pass filter used to block the excitation laser.

Single-photon emission purity is calculated by fitting the autocorrelation histogram under pulsed excitation with Lorentzian functions. The value of $g^{(2)}(0)$ is calculated dividing the area of the peak at zero delay time by the average area of 10 other peaks at $\tau \neq 0$.

**Simulations**. A COMSOL Multiphysics simulation is conducted to calculate strain in various directions of the substrate. Density, Young's modulus, and Poisson's ratio of polycarbonate substrate are set to 1.21 g m$^{-3}$, 2.2 GPa and 0.37, respectively. Substrate size of $100 \times 12 \times 1.5$ mm is used similarly to our experimental setup. The left plane of the PC beam is fixed and a vertical uniform force is loaded along a line at 40 mm away from the clamped edge.

All calculations for the energy shift as a function of strain are undertaken using the SIESTA[42] implementation of density functional theory with the Perdew Burke Ernzerhof[41] approximation to the exchange-correlation functional. The nucleus–electron interaction is represented by norm-conserving pseudopotentials calculated according to the method described by Troullier and Martins[43]. Electronic charge is represented by numerical pseudo-atomic orbitals equivalent to a double-zeta plus polarization basis set. In SIESTA, these orbitals are strictly confined to a cutoff radius determined by a single energy value representing the shift in orbital energy due to confinement. Here a confinement energy of 5 mRy is used. This value ensures sufficient convergence without excessive computational time. Pristine single-layer hBN is first geometrically optimized using the conventional cell and a $21 \times 21 \times 1$ Monkhurst-Pack reciprocal space grid to a tolerance of 0.01 eV Å$^{-1}$. A large vacuum of 30 Å is used to ensure that interaction between periodic images is negligible. The optimized lattice parameter is 2.5 Å

with a bond length of 1.452 Å. To avoid off-diagonal elements (shear strain) in the strain tensor and to ensure that the interaction between $N_B V_N$ defect and its images is negligible in the direction of the optical dipole, two different supercells are used, i.e., for simulating biaxial strain and strain along ZZ1, AC2, ZZ1-P and AC2-P, a $9 \times 11 \times 1$ $(7 \times 7 \times 1)$ supercell is used. All the defect structures were re-optimized by allowing the atoms to relax to a force tolerance of/or better than $0.04 \, eV \, Å^{-1}$. All calculations are spin-polarized. A real space integration grid with an equivalent plane-wave cutoff of 750 Ry is used and is sufficient to ensure numerical convergence. The imaginary component of the frequency dependent dielectric matrix is calculated in the Random Phase Approximation using an optical mesh of $14 \times 14 \times 1$ and a Gaussian broadening of 0.06 eV.

**Data availability**. The data that support the findings of this study are available from the authors on reasonable request, see author contributions for specific data sets.

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

## Acknowledgements

This work was supported in part by the Center for Excitonics, an Energy Frontier Research Center funded by the US Department of Energy, Office of Science, Office of Basic Energy Sciences under award no. DE-SC0001088. Measurements were supported in part by the National Science Foundation EFRI 2-DARE, award abstract no. 1542863, in part by the US Army Research Laboratory (ARL) program Center For Distributed Quantum Information, and the CIQM (grant DMR-1231319). G.G. acknowledges support by the Swiss National Science Foundation (SNSF). H.M. is a recipient of the Samsung Scholarship. B.L. acknowledges financial funding by CIQM & CDQI. M.M.F. acknowledges financial funding by the Austrian Science Fund (START Y-539). I.A. acknowledges the financial support from the Australian Research Council (project no. DE130100592).

## Author contributions

G.G. and H.M. developed and carried out the sample processing, performed optical measurements and analyzed the data. B.L. performed AFM measurements, assisted with sample processing and optical measurements. G.G. and H.M. conceived and developed the strain tuning apparatus. D.K.E., M.M.F. and P.J.H. assisted with the sample transfer and participated to discussions. S.A. and M.J.F. performed simulations. G.G., H.M., I.A.

and D.E. wrote the paper. I.A. and D.E. conceived and directed the project. All authors discussed the results and commented on the manuscript.

## Additional information

**Competing interests:** The authors declare no competing financial interests.

