## [Peer Review file · Nature Communications]

Editorial Note: This manuscript has been previously reviewed at another journal that is not operating a transparent peer review scheme. This document only contains reviewer comments and rebuttal letters for versions considered at Nature Communications. Mentions of prior referee reports have been redacted.

Reviewers' comments:

Reviewer #1 (Remarks to the Author):

The authors have basically addressed my previous comments.

Reviewer #2 (Remarks to the Author):

I appreciate the authors' efforts in addressing my concerns and in modifying the manuscript. After reading through the replies, I still have one question regarding the strain effects. The authors argue that the applied strain is not the same as the actual strain on the defect centers, which I find reasonable. However, the authors claim the strain varies with the layer thickness as $(t)^N$ without providing any details. More details of the model and the derivations should be provided. I would support the publication of the manuscript in Nature Communications if this point could be addressed.

Reviewer #3 (Remarks to the Author):

The manuscript by Grosso et al presents an investigation of room temperature single photon emission from crystal defects in hBN. There are two novel results in the manuscript: (1) using material processing to achieve improved single photon purity due to reduced background fluorescence; and (2) strain tuning the energy of the emitted single photons over a 6 meV range.

Unfortunately, I do not believe this work is suitable for publication in Nature Communications for the following reasons:

(1) The prospects of hBN atomic defects in quantum information processing devices is oversold in the manuscript. For instance, unlike diamond defects there is no spin to speak of hBN atomic defects (to date at least), and unlike other single photon sources such as III-V quantum dots the zero phonon line is small compared to the phonon sideband, signifying that there is little prospect for generating high quality indistinguishable photons. In fact, the authors state that the strain tuning "increases the probability of spectral overlapping different emitters" energy – but it's not clear why this is actually useful as there is little chance they could be made to be indistinguishable as they are so incoherent (even at low temperature).

(2) The strain tuning result is a nice first result but still quite far from the state of the art. Strain tuning is of course expected to work – and work very well – for 2D materials because they should be easier to strain and can be strained more than bulk materials. Nevertheless, the results achieved here do not compare well with what can be done in bulk semiconductors [e.g. strain tuning of GaAs quantum dots by Trotta / Rastelli et al]. Again, the usefulness of this strain tuning is stated to be for spectrally overlapping multiple emitters, but the prospects for obtaining indistinguishability seems to be oversold.

(3) The material processing to improve single photon purity is a nice result. But it still isn't so 'pure' – 8% in the best case (low excitation power). Even worse, at higher excitation powers it is still necessary to correct for an uncorrelated photon background. Again, other quantum emitter platforms have shown much better single photon purity and can be truly isolated so that background subtraction is not required.

More specific comments/suggestions:

- along with ref 16 I believe the state of the art in strain tuning QDs should be cited too [e.g. Trotta / Rastelli]
- the recent works on strain tuning in monolayer WSe₂ by the Atature and Gerardot groups should be cited at the top of page 3
- please state the filter bandwidth that is used to obtain the $g_2(0) = 7.7\%$
- top of page 5: please state that the saturation curve is performed with CW excitation

- the strain tuning curves are quite noisy relative to the linear fit. How reversible are they? Is there significant hysteresis?
- how is the crystal lattice aligned to the cantilever?
- the y-axis label for the g_2 curve in fig 1a is wrong [should be "normalized" or just $g_2(\tau)$]- but I do think it would be nice to show the coincidence counts on the right y-axis too.
- Fig 1 caption: would be good to mention the experiment temperature.
- Fig 2 caption: at the very end it would be good to state the fraction of light in the ZPL.
- it would be useful to include the lifetime measurement (Fig 10S) in the main manuscript.

Reviewer #1 (Remarks to the Author):

The authors have basically addressed my previous comments.

Reply: We thank the reviewer #1 for the favourable opinion of our work and for appreciating our efforts. We are particularly grateful for the first round of comments, which allowed us to improve the quality of the manuscript.

Reviewer #2 (Remarks to the Author):

I appreciate the authors' efforts in addressing my concerns and in modifying the manuscript. After reading through the replies, I still have one question regarding the strain effects. The authors argue that the applied strain is not the same as the actual strain on the defect centers, which I find reasonable. However, the authors claim the strain varies with the layer thickness as $(t)^N$ without providing any details. More details of the model and the derivations should be provided. I would support the publication of the manuscript in Nature Communications if this point could be addressed.

Reply: We would like to thank the referee for the detailed review of our work, which has raised many illuminating points that have improved the quality of our manuscript. We also appreciate this further question that highlights an unclear passage in the new version of the manuscript. We answer this question in detail in the following. We highlight in blue the parts of the text that have been modified/added with respect of previous versions of the manuscript.

The equation for the transmitted strain among layers is based on the simulation results reported in Ref.36 of the manuscript. In this work, the strain transmitted among two consecutive layers of a bilayer material is found to be $t \sim 98\%$. Due to the weak van der Waals bound among hBN layers, we assume that this correction factor t holds also for all the other layers of thick flakes. In our case, this assumption implies that the bottom layer of the thick hBN flake (the one in contact with the top surface of the bendable substrate) experiences an effective strain equal to $S_{eff}^{(0)} = S_{app}$, where S_{app} is the strain applied to the substrate. The second layer of the hBN flake experiences an effective strain equals to $S_{eff}^{(1)} = S_{eff}^{(0)} \cdot t = S_{app} \cdot t$. The third layer experiences a strain of $S_{eff}^{(2)} = S_{eff}^{(1)} \cdot t = (S_{app} \cdot t) \cdot t$, and so on. The n -th layer of the flake experiences a strain of $S_{eff}^{(n)} = S_{eff}^{(n-1)} \cdot t = (S_{app} \cdot t^{n-1}) \cdot t = S_{app} \cdot t^n$. Therefore, the effective strain on the n -th layer scales as $\sim t^n$. For the strain experiments, we used hBN flakes of thickness $\sim 50\text{nm}$, which gives an estimation for the number of layers $N \sim 150$.

To make this model clear in the text we change the following sentence:

“Considering an average of $N \sim 150$ layers from the substrate to the layer of the emitter, we have that $S_{eff} = S_{app} \cdot t^N = S_{app} \cdot 0.05$, where S_{eff} is the effective strain at the emitter, S_{app} the applied strain.”

with:

“We assume that in thicker flakes, the correction factor for transmitted strain t is the same for all consecutive layers. Therefore, the effective strain at the emitter location (S_{eff}) is given by the strain applied to the bottom layer of the hBN flake (S_{app}) scaled by the transmitted strain t among the N layers separating the emitter and the substrate, and reads

$S_{eff}^N = S_{eff}^{N-1} \cdot t = (S_{eff}^{N-2} \cdot t) \cdot t = \dots = S_{eff}^1 \cdot t^{N-1} = S_{app} \cdot t^N$. Considering an average of $N \sim 150$ layers for our experiments, we obtain $S_{eff} = S_{app} \cdot t^N = S_{app} \cdot 0.05$.”

Reviewer #3 (Remarks to the Author):

The manuscript by Grosso et al presents an investigation of room temperature single photon emission from crystal defects in hBN. There are two novel results in the manuscript: (1) using material processing to achieve improved single photon purity due to reduced background fluorescence; and (2) strain tuning the energy of the emitted single photons over a 6 meV range.

Unfortunately, I do not believe this work is suitable for publication in Nature Communications for the following reasons:

(1) The prospects of hBN atomic defects in quantum information processing devices is oversold in the manuscript. For instance, unlike diamond defects there is no spin to speak of hBN atomic defects (to date at least), and unlike other single photon sources such as III-V quantum dots the zero phonon line is small compared to the phonon sideband, signifying that there is little prospect for generating high quality indistinguishable photons. In fact, the authors state that the strain tuning “increases the probability of spectral overlapping different emitters” energy – but it’s not clear why this is actually useful as there is little chance they could be made to be indistinguishable as they are so incoherent (even at low temperature).

(2) The strain tuning result is a nice first result but still quite far from the state of the art. Strain tuning is of course expected to work – and work very well – for 2D materials because they should be easier to strain and can be strained more than bulk materials. Nevertheless, the results achieved here do not compare well with what can be done in bulk semiconductors [e.g. strain tuning of GaAs quantum dots by Trotta / Rastelli et al]. Again, the usefulness of this strain tuning is stated to be for spectrally overlapping multiple emitters, but the prospects for obtaining indistinguishability seems to be oversold.

(3) The material processing to improve single photon purity is a nice result. But it still isn’t so ‘pure’ – 8% in the best case (low excitation power). Even worse, at higher excitation powers it is still necessary to correct for an uncorrelated photon background. Again, other quantum emitter platforms have shown much better single photon purity and can be truly isolated so that background subtraction is not required.

Reply: We thank the referee for the detailed review of our work and for the many insightful comments provided. While we appreciate that the reviewer acknowledges the novelties of our findings, we would like to stress that the scope of our work is not, at the present stage, to provide a competitive alternative to the state-of-the-art in single photon emitters and atom-like defects, but rather to discuss a promising and emerging system. We are aware that, at the current stage, hBN still presents technical limitations for applications in quantum information processing. Nevertheless, we believe that

the exceptional properties of 2D materials combined with the promising experimental results on atomic-defects present an encouraging starting point for understanding quantum properties of atomically layered structures and for developing the next generation of host materials. For these reasons, and from our discussions with colleagues, we think that these early studies on atom-like emitters in low-dimensional materials are of great interest for the scientific community.

We would like to highlight that atom-like defects in hBN are relatively newcomers to the field of quantum emitters and, at this point, we do not know whether the photons are coherent or not at low temperature, and whether they are indistinguishable. Compared to other systems, the brightness of hBN is favorable, and so is the availability of wavelength selection. These properties are promising for practical devices - for instance for coupling to cavities or plasmonic resonators, as it has been already demonstrated. Therefore, we do not think that the prospects of hBN are oversold.

We should also mention that the strain tuning in hBN is comparable with color centers in other solids and also larger than other 2D TMDs. The paper that the reviewer refers to is indeed very interesting, but requires a very specific QD in a very specific design. Our systems are complementary, and we do not aim to compete with other bulk materials. As mentioned earlier, applications may be different and there is great scientific interest in exploring the newly found defects in hBN.

At the same time, we do appreciate the concerns by the reviewer that this emitter must be viewed in the appropriate context. To this end, we added to the manuscript the references suggested by the reviewer and modified certain passages of the text in order to help the reader to better contextualize our findings. Please find below a detailed list of modifications.

More specific comments/suggestions:

We are grateful to the reviewer for raising many interesting and fruitful technical questions that have surely helped to increase the overall quality of our work. We address each point individually, highlighting in blue the parts of the text that have been modified/added with respect to the previous version of the manuscript.

- along with ref 16 I believe the state of the art in strain tuning QDs should be cited too [e.g. Trotta / Rastelli]

Reply: As suggested by the reviewer, we added the reference “Trotta, R. & Rastelli, A. in *Engineering the Atom-Photon Interaction* 277–302 (2015)” to the last sentence of the first subsection of the “Results”, that we changed from:

“Thus, external strain promises a particularly effective method to control the optical properties of embedded quantum emitters in 2D materials, as we will show below. ”

To:

“Similarly to other solid-state systems^{23,24}, external strain promises a particularly effective method to control the optical properties of embedded quantum emitters in 2D materials, as we will show below.”

- the recent works on strain tuning in monolayer WSe2 by the Atature and Gerardot groups should be cited at the top of page 3

Reply: We thank the reviewer for this useful suggestion. When we submitted our manuscript for the first time, both papers just appeared on preprint servers and they are still on the arXiv. Nevertheless, we agree that these nice and similar works should be cited and we add them to the main text. We added Ref.20, 21, 22.

On page 3 we modified the following sentences:

“The unusually high stretchability of 2D materials allows for effective strain engineering of physical and optical properties^{15,16}, including giant tunability of the electronic bandgap. The strain-induced displacement Δa of lattice sites deforms the molecular orbitals of the atom-like defects and perturbs their energy levels according to...”

With:

“The unusually high stretchability of 2D materials allows for effective strain engineering of physical and optical properties^{15,16}, including giant tunability of the electronic bandgap. **Recently, it has been shown that local strain in TMDs can be used to confine individual excitons with promising properties for single photon applications¹⁷.** In the case of atom-like defects, the strain-induced displacement Δa of lattice sites deforms the molecular orbitals and perturbs their energy levels according to...”

- please state the filter bandwidth that is used to obtain the $g^2(0) = 7.7\%$

Reply: In our experiments we did not use any spectral filtering for the measurement of the second order correlation function, except for a 550nm long pass filter to block the excitation laser at 532nm. To make this important aspect clear in the the text, we added the following sentence to the manuscript when discussing Fig.3a:

“Fig.3a plots the second-order correlation histogram ($g^{(2)}(\tau)$) of a typical emitter from the irradiated area, as measured **integrating over the entire emission spectrum and** using a Hanbury Brown and Twiss setup with pulsed excitation of 300 ps and 20 MHz repetition rate.”

We also added to the Methods section a sentence to clarify the experimental conditions:

“**Second-order correlation histogram is measured integrating over the entire emission spectrum and without any spectral filtering, except for the 550 nm long pass filter used to block the excitation laser.**”

- top of page 5: please state that the saturation curve is performed with CW excitation

Reply: We changed the sentence discussing the saturation curve in Fig.3c, from:

“Fig.3c plots the rate of collected photons from a single hBN emitter as a function of excitation power.”

To:

“**Fig.3c plots the rate of collected photons from a single hBN emitter as a function of continuous wave (CW) excitation power.**”

- the strain tuning curves are quite noisy relative to the linear fit. How reversible are they? Is there significant hysteresis?

Reply: Several experimental results have been reported in the literature on the effects of strain in various 2D materials, including mono- and multi-layer MoS₂, graphene/hBN stacks, bilayer phosphorene. In all these experiments, elastic strain tuning has been observed with no hysteresis within the strain range of -2% to 2% (Ref.37) even for multilayers. In particular, for the case of trilayer MoS₂ it has been observed that “*PL peak position follow the path very well even after several months of repeated measurements*” (Supplementary Information of Ref.38). In other experiments in graphene/hBN stack, this limit can range up to 10-20% (Ref.39 and Ref.40).

In our experiments, the maximum strain applied to the bottom layer of the hBN flakes is 1%, thus within the elasticity limits. Moreover, the effective strain at the emitter location needs to be scale due to the thickness of the flake, resulting in a much smaller strain with negligible memory effects on the deformation of the atom-like defect.

We make this point clear in the text when discussing the experimental results by adding a sentence on page 7 with all the references mentioned above:

“The exceptional elasticity of two-dimensional materials suggests that the tuning of SPEs in hBN is a reversible process within the strain range used in our experiments³⁰⁻³³ .”

- how is the crystal lattice aligned to the cantilever?

Reply: The reviewer’s question is probably related to the angle between the strain axis (parallel to the cantilever) and the orientation of the emitter dipole. An experimental estimation of this angle is extremely arduous because the orientation of the dipole seems to be only weakly correlated to the crystallographic axis, as recently reported by the work of Prof. Bassett’s group, cited in Ref.15. We agree with the reviewer that this is indeed an important aspect of our experiments. For this reason, we have discussed theoretically the role of this angle relative to the different crystal orientations in the last paragraph of the main text. The simulated effects of the strain along different crystal orientations are reported in Supplementary Fig.14, in which it is possible to notice the effects of strain in the emission energy of the quantum emitters as a function of the strain-dipole angle.

- the y-axis label for the g_2 curve in fig 1a is wrong [should be “normalized” or just $g_2(\tau)$]- but I do think it would be nice to show the coincidence counts on the right y-axis too.

Reply: Thanks for pointing out this mistake. We have changed the label of Fig.1a from “Coincident Counts” to “ $g^{(2)}(\tau)$ ”. The coincidence counts for the second order correlation function with pulsed excitation are shown in Fig.3a and Fig.3b. The scope of Fig.1a is to show the antibunched light emitted by the atomic defects and we think this is already clear from the normalized counts. On the other hand, we think that showing the coincidence counts on the right axis could make Figure 1 too crowded.

- Fig 1 caption: would be good to mention the experiment temperature.

Reply: Despite the title of the manuscript is already quite clear regarding the operation temperature of single photon emitters in hBN, we agree with the reviewer that it is useful to stress this crucial advantage of this system also in the explicative figure.

We modified the title of the caption of Figure 1 from:

”FIG. 1: **Single photon emission of atom-like defect in hexagonal boron nitride:**”

to:

“**FIG. 1: Single photon emission of atom-like defects in hexagonal boron nitride at room temperature:**”

- Fig 2 caption: at the very end it would be good to state the fraction of light in the ZPL.

Reply: Similarly to Ref.15, we estimated the fraction of light in the ZPL with a simple fit of the emission spectrum assuming a Lorentzian shape for the ZPL and the phonon side bands. From our results we obtain a spectral weight for the ZPL of ~30%.

We modified the last line of the caption for Fig.2 in the following way:

“This emitter shows a zero phonon line (ZPL) emission at 2.145 eV **with a spectral weight of ~30%.**”

We also added Ref.15 to the introduction.

- it would be useful to include the lifetime measurement (Fig 10S) in the main manuscript.

Reply: Lifetime measurements of the same emitter and in similar experimental conditions have been already reported in several publications, including Ref.13 and Ref.18 of the main text. Our results are in excellent agreement with previous reports (confirming the ~3ns lifetime for this particular emitter) and do not represent any significant progress in the study of this system.

Therefore, we believe that showing the results in the main text will not add any relevant piece of information to the field and, for sake of space and clarity, we prefer to leave this result in the Supplementary Information. Nevertheless, we agree with the reviewer that this piece of information is missing in the manuscript and it should be added to the main text, with a reference to Supplementary Fig.10.

On page 5 we added the following sentence:

“**Time-dependent measurements revealed a lifetime of 2.4 ns for this emitter (see Supplementary Note 2 and Supplementary Fig.10), in excellent agreement with previous reports^{13,18}.**”

REVIEWERS' COMMENTS:**Reviewer #2 (Remarks to the Author):**

The authors have addressed my question and I have no further comments. Although the impact of the work on quantum information processing still needs to be seen, the work has shown certain new and interesting aspects of hBN atomic defects and could be published in Nature Communications.

Reviewer #3 (Remarks to the Author):

The authors have suitably addressed the suggestions in the previous refereeing round. The results are high quality and technically correct. I do disagree about the prospects for generation of photons with a high degree indistinguishability in this platform: if the zero-phonon fraction is only 30% of the total emitted light the prospects do not look great.

REVIEWERS' COMMENTS:

Reviewer #2 (Remarks to the Author):

The authors have addressed my question and I have no further comments. Although the impact of the work on quantum information processing still needs to be seen, the work has shown certain new and interesting aspects of hBN atomic defects and could be published in Nature Communications.

We thank Reviewer #2 for the positive opinion of our work, the helpful comments provided during the review process, and for recommending publication to Nature Communications.

Reviewer #3 (Remarks to the Author):

The authors have suitably addressed the suggestions in the previous refereeing round. The results are high quality and technically correct. I do disagree about the prospects for generation of photons with a high degree indistinguishability in this platform: if the zero-phonon fraction is only 30% of the total emitted light the prospects do not look great.

We thank Reviewer #3 for the positive opinion of our work and the helpful comments provided during the review process. We agree with Reviewer #3 that hBN system still presents limitations for indistinguishability and other requirements for quantum information processing. Nevertheless, we believe that some of these limitations could be overcome in the future with new material processing and further investigations.